# A biophysical threshold for biofilm formation

Jenna A Moore-Ott[1], Selena Chiu[1], Daniel B Amchin[1], Tapomoy Bhattacharjee[2], Sujit S Datta[1]*

[1]Department of Chemical and Biological Engineering, Princeton University, Princeton, United States; [2]Andlinger Center for Energy and the Environment, Princeton University, Princeton, United States

**Abstract** Bacteria are ubiquitous in our daily lives, either as motile planktonic cells or as immobilized surface-attached biofilms. These different phenotypic states play key roles in agriculture, environment, industry, and medicine; hence, it is critically important to be able to predict the conditions under which bacteria transition from one state to the other. Unfortunately, these transitions depend on a dizzyingly complex array of factors that are determined by the intrinsic properties of the individual cells as well as those of their surrounding environments, and are thus challenging to describe. To address this issue, here, we develop a generally-applicable biophysical model of the interplay between motility-mediated dispersal and biofilm formation under positive quorum sensing control. Using this model, we establish a universal rule predicting how the onset and extent of biofilm formation depend collectively on cell concentration and motility, nutrient diffusion and consumption, chemotactic sensing, and autoinducer production. Our work thus provides a key step toward quantitatively predicting and controlling biofilm formation in diverse and complex settings.

## Editor's evaluation

In this work, the authors develop a continuum description of biofilm formation from initially planktonic cells. The coupled partial differential equations that encode the dynamics of the cell populations, nutrients and autoinducers contain many parameters, but it is shown that only two dimensionless combinations of them are needed to understand the threshold for biofilm formation. This work should be of broad interest to a wide range of researchers in biophysics and cell biology.

**\*For correspondence:**
ssdatta@princeton.edu

**Competing interest:** The authors declare that no competing interests exist.

## Introduction

Dating back to their discovery by van Leeuwenhoek over three centuries ago, it has been known that bacteria typically exist in one of two phenotypic states: either as motile, planktonic cells that self-propel using e.g., flagella or pili ("animalcules … moving among one another"; *Van Leewenhoeck, 1677*), or as immobilized, surface-attached biofilms ("little white matter … in the scurf of the teeth"; *Leewenhoeck, 1684*). These different states have critical functional implications for processes in agriculture, environment, industry, and medicine. For example, motility-mediated dispersal of planktonic cells enables populations to escape from harmful conditions and colonize new terrain (*Adler, 1966a*; *Adler, 1966b*; *Saragosti et al., 2011*; *Fu et al., 2018*; *Cremer et al., 2019*; *Bhattacharjee et al., 2021*)—underlying infection progression, drug delivery to hard-to-reach spots in the body, food spoilage, interactions with plant roots in agriculture, and bioremediation of environmental contaminants (*Balzan et al., 2007*; *Chaban et al., 2015*; *Datta et al., 2016*; *Harman et al., 2012*; *Ribet and Cossart, 2015*; *Siitonen and Nurminen, 1992*; *Lux et al., 2001*; *O'Neil and Marquis, 2006*; *Gill and Penney, 1977*; *Shirai et al., 2017*; *Thornlow et al., 2015*;

*Toley and Forbes, 2012*; *Dechesne et al., 2010*; *Souza et al., 2015*; *Turnbull et al., 2001*; *Watt et al., 2006*; *Babalola, 2010*; *Adadevoh et al., 2016*; *Adadevoh et al., 2018*; *Ford and Harvey, 2007*; *Wang et al., 2008*; *Reddy and Ford, 1996*; *Martínez-Calvo et al., 2021*). In addition, the formation of immobilized biofilms can initiate antibiotic-resistant infections, foul biomedical devices and industrial equipment, or conversely, help sequester and remove contaminants in dirty water (*Davey and O'toole, 2000*; *Hall-Stoodley et al., 2004*; *Mah et al., 2003*; *O'Toole and Stewart, 2005*; *Fux et al., 2005*; *Nicolella et al., 2000*; *Donlan and Costerton, 2002*; *Davies et al., 1998*). Hence, extensive research has focused on understanding bacterial behavior in either the planktonic or biofilm state.

For example, studies of planktonic cells have provided important insights into bacterial motility—which can be either undirected (*Berg, 2018*; *Berg, 2004*; *Bhattacharjee and Datta, 2019a*; *Bhattacharjee and Datta, 2019b*) or directed in response to e.g., a chemical gradient via chemotaxis (*Adler, 1966b*; *Adler, 1966a*; *Saragosti et al., 2011*; *Fu et al., 2018*; *Cremer et al., 2019*; *Bhattacharjee et al., 2021*; *Keller and Segel, 1971*; *Odell and Keller, 1976*; *Keller and Odell, 1975*; *Lauffenburger, 1991*; *Seyrich et al., 2019*; *Croze et al., 2011*; *Amchin et al., 2022*). These processes are now known to be regulated not just by intrinsic cellular properties, such as swimming kinematics and the amplitude and frequency of cell body reorientations, but also by the properties of their environment, such as cellular concentration, chemical/nutrient conditions, and confinement by surrounding obstacles (*Berg, 2018*; *Berg, 2004*; *Bhattacharjee and Datta, 2019a*; *Bhattacharjee and Datta, 2019b*; *Adler, 1966b*; *Adler, 1966a*; *Saragosti et al., 2011*; *Fu et al., 2018*; *Cremer et al., 2019*; *Bhattacharjee et al., 2021*; *Keller and Segel, 1971*; *Odell and Keller, 1976*; *Keller and Odell, 1975*; *Lauffenburger, 1991*; *Seyrich et al., 2019*; *Croze et al., 2011*; *Amchin et al., 2022*). Thus, the manner in which planktonic bacteria disperse can strongly vary between different species and environmental conditions.

Similarly, studies of biofilms under defined laboratory conditions have also provided key insights—such as by revealing the pivotal role of intercellular chemical signaling in biofilm formation (*Nadell et al., 2008*; *Bassler and Losick, 2006*; *Davey and O'toole, 2000*; *Hall-Stoodley et al., 2004*). In this process, termed quorum sensing, individual cells produce, secrete, and sense freely diffusible autoinducer molecules, thereby enabling different bacteria to coordinate their behavior (*Davies et al., 1998*; *Sakuragi and Kolter, 2007*; *Bassler and Losick, 2006*; *Miller and Bassler, 2001*; *Herzberg et al., 2006*; *Laganenka et al., 2018*; *McLean et al., 1997*; *Paul et al., 2009*). For example, in many cases, quorum sensing positively controls biofilm formation (*Herzberg et al., 2006*; *Laganenka et al., 2018*; *Davies et al., 1998*; *Sakuragi and Kolter, 2007*; *McLean et al., 1997*; *González Barrios et al., 2006*; *Yarwood et al., 2004*; *Koutsoudis et al., 2006*; *Waters and Bassler, 2005*; *Parsek and Greenberg, 2005*; *Jayaraman and Wood, 2008*; *Hentzer et al., 2003*; *Kirisits and Parsek, 2006*): autoinducer accumulation above a threshold concentration upregulates the expression of genes involved in biofilm formation, ultimately driving a transition from the planktonic to the biofilm state (*Nadell et al., 2008*). Again, however, the cellular factors that control this transition, such as the autoinducer production rate, diffusivity, and threshold concentration, can strongly vary between different species and environmental conditions.

Because planktonic dispersal and biofilm formation both depend on a dizzyingly complex array of factors, these distinct processes are typically studied in isolation. Thus, while each is well understood on its own, quantitative prediction of the conditions under which a population of planktonic bacteria transitions to the biofilm state—or instead, continues to disperse away and remains in the planktonic state—remains challenging. Here, we address this challenge by developing a mathematical model that describes essential features of motility-mediated dispersal of planktonic cells and autoinducer-mediated biofilm formation together. Using numerical simulations of this model, we systematically examine the influence of cellular concentration, motility, and chemotactic sensing; nutrient availability, diffusion, and consumption; and autoinducer production, diffusion, and accumulation on biofilm formation. Guided by these results, we establish a potentially-universal biophysical threshold that unifies the influence of all these factors in predicting the onset and extent of biofilm formation across different species and environmental conditions. Our work therefore provides a theoretical foundation for the prediction and control of biofilm formation in diverse and complex settings, and yields new quantitative predictions to guide future experiments.

## Results

### Development of the governing equations

As an illustrative example, and to connect our model to recent experiments of bacterial dispersal (*Bhattacharjee et al., 2021*), we consider a rectilinear geometry with a starting inoculum of planktonic cells at a maximal concentration $b_{1,0}$ and of width $x_0$. In general, the continuum variable $b(x, t)$ describes the number concentration of bacteria, where $x$ is the position coordinate and $t$ is time, and the subscripts $\{1, 2\}$ represent planktonic or biofilm-associated cells, respectively. Following previous work (*Lauffenburger, 1991*; *Keller and Segel, 1971*; *Adler, 1966a*; *Croze et al., 2011*; *Fu et al., 2018*; *Bhattacharjee et al., 2021*), we consider a sole diffusible nutrient that also acts as

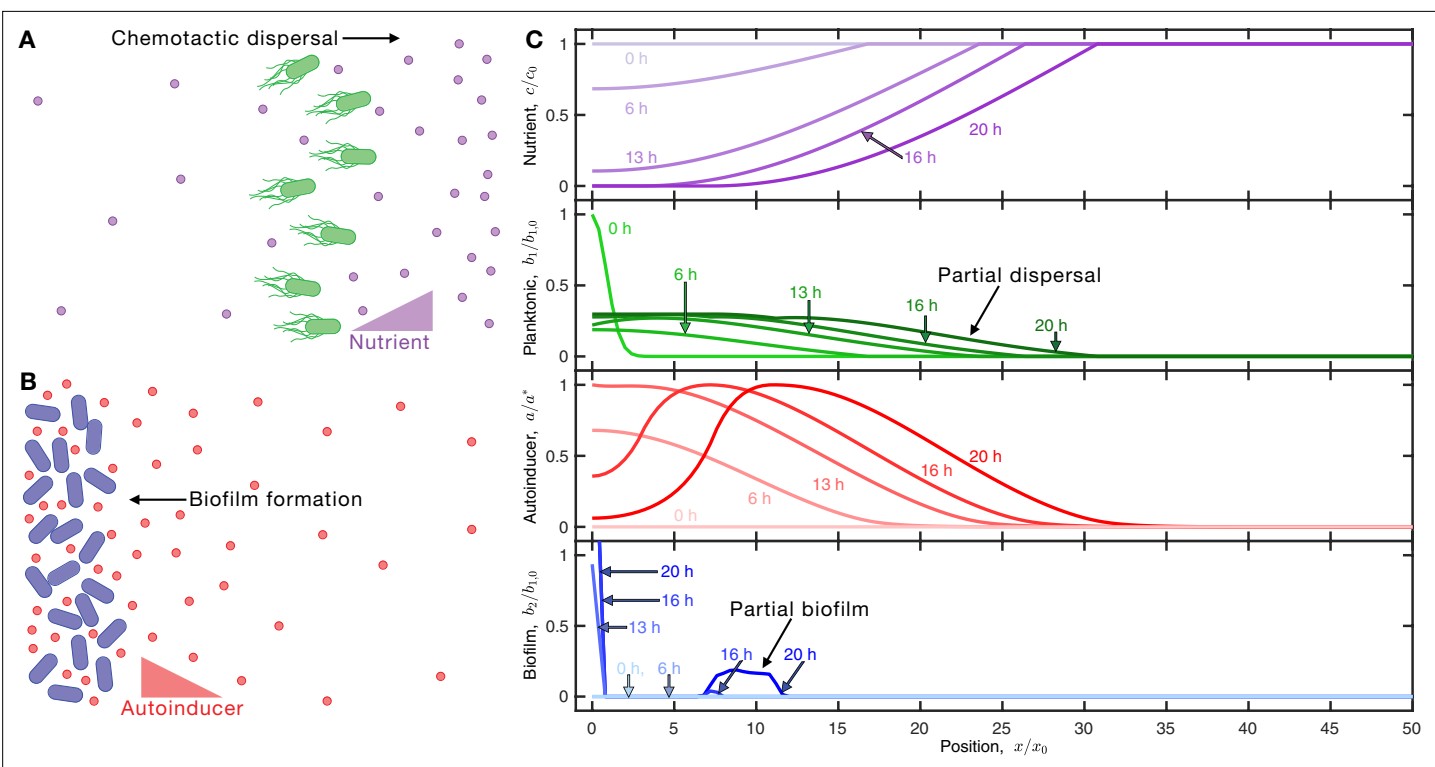

**Figure 1.** Competition between motility-mediated dispersal and autoinducer-mediated biofilm formation. (**A**) Schematic of chemotactic dispersal: planktonic bacteria (green) consume nutrient (purple) and establish a local gradient that they, in turn, direct their motion in response to. (**B**) Schematic of positive quorum sensing-controlled biofilm formation: accumulation of produced autoinducer (red) above a threshold concentration causes cells to transition to the biofilm state (blue). (**C**) Results of an example simulation of *Equations 1–4* showing the dynamics of the nutrient, planktonic cells, autoinducer, and biofilm cells from top to bottom, quantified by the normalized concentrations $c/c_0$, $b_1/b_{1,0}$, $a/a^*$, $b_2/b_{1,0}$, respectively; $c_0$, $b_{1,0}$, and $a^*$ represent the initial nutrient concentration, initial bacterial concentration, and autoinducer threshold for biofilm formation, respectively. The position coordinate is represented by the normalized position $x/x_0$, where $x_0$ is the width of the initial cellular inoculum. Different shades indicate different time points as listed. The inoculum initially centered about the origin consumes nutrient (purple), establishing a gradient that drives outward dispersal by chemotaxis (outward moving green curves); the cells also produce autoinducer (red) concomitantly. At $t \approx 13\,\mathrm{h}$, sufficient autoinducer has been produced to trigger biofilm formation at the origin; at even longer times ($t \gtrsim 16\,\mathrm{h}$), nutrient depletion limits autoinducer production at this position. However, accumulation of autoinducer by the dispersing planktonic cells triggers partial biofilm formation at $x/x_0 \approx 6$ as well. This competition between dispersal and biofilm formation leads to a final biofilm fraction of $f = 21\%$ at the final time of $t = 20\,\mathrm{h}$. An animated form of this figure is shown in *Video 1*. The values of the simulation parameters are given in *Supplementary file 2*.

The online version of this article includes the following source data and figure supplement(s) for figure 1:

**Source data 1.**

**Figure supplement 1.** The exact nature of the temporal dynamics of the arrest in motility while transitioning to the biofilm state does not appreciably influence our model results.

**Figure supplement 1—source data 1.**

**Figure supplement 2.** The details of the initial inoculum shape do not appreciably influence our model results.

**Figure supplement 2—source data 1.**

the chemoattractant, with a number concentration represented by the continuum variable $c(x, t)$ with diffusivity $D_c$. Initially, nutrient is replete throughout the system at a constant concentration $c_0$. The bacteria then consume the nutrient at a rate $b_1 \kappa_1 g(c)$, where $\kappa_1$ is the maximum consumption rate per cell and the Michaelis-Menten function $g(c) \equiv \frac{c}{c + c_{\text{char}}}$ quantifies the nutrient dependence of consumption relative to the characteristic concentration $c_{\text{char}}$ (*Croze et al., 2011*; *Monod, 1949*; *Cremer et al., 2019*; *Woodward et al., 1995*; *Shehata and Marr, 1971*; *Schellenberg and Furlong, 1977*; *Cremer et al., 2016*).

As time progresses, the bacteria thereby establish a local nutrient gradient that they respond to via chemotaxis (*Figure 1A*). In particular, planktonic cells disperse through two processes: undirected active diffusion with a constant diffusivity $D_1$ (*Berg, 2018*), and directed chemotaxis with a drift velocity $\vec{v}_c \equiv \chi_1 \nabla \log \left( \frac{1 + c/c_-}{1 + c/c_+} \right)$ that quantifies the ability of the bacteria to sense and respond to the local nutrient gradient (*Keller and Segel, 1970*; *Keller and Segel, 1971*; *Odell and Keller, 1976*; *Keller and Odell, 1975*) with characteristic bounds $c_-$ and $c_+$ (*Cremer et al., 2019*; *Sourjik and Wingreen, 2012*; *Shimizu et al., 2010*; *Tu et al., 2008*; *Kalinin et al., 2009*; *Shoval et al., 2010*; *Lazova et al., 2011*; *Celani et al., 2011*; *Fu et al., 2018*; *Dufour et al., 2014*; *Yang et al., 2015*; *Cai et al., 2016*; *Chen and Jin, 2011*) and a chemotactic coefficient $\chi_1$. The planktonic cells also proliferate at a rate $b \gamma_1 g(c)$, where $\gamma_1$ is the maximal proliferation rate per cell. Finally, as the planktonic bacteria consume nutrients, they produce and secrete a diffusible autoinducer, with a number concentration represented by $a(x, t)$ and with diffusivity $D_a$, at a maximal rate $k_1$ per cell. Motivated by some previous work (*Hense et al., 2012*; *Hense and Schuster, 2015*; *Kirisits et al., 2007*; *Bollinger et al., 2001*; *Duan and Surette, 2007*; *Mellbye and Schuster, 2014*; *De Kievit et al., 2001*; *Pérez-Osorio et al., 2010*), we take this process (hereafter referred to as 'production' for brevity) to also be nutrient-dependent via the same Michaelis-Menten function $g(c)$ for the results presented in the main text, but we also consider the alternate case of 'protected' nutrient-independent production in the supplementary materials. Following previous work (*Koerber et al., 2002*; *Ward et al., 2001*; *Ward et al., 2003*), we also model natural degradation of autoinducer as a first-order process with a rate constant $\lambda$.

As autoinducer is produced, it binds to receptors on the surfaces of the planktonic cells with a second-order rate constant $\alpha$, as established previously (*Koerber et al., 2002*; *Ward et al., 2001*; *Ward et al., 2003*). Motivated by experiments on diverse bacteria, including the prominent and well-studied species *Escherichia coli*, *Pseudomonas putida*, and *Pseudomonas aeruginosa* (*Davies et al., 1998*; *Sakuragi and Kolter, 2007*; *Bassler and Losick, 2006*; *Miller and Bassler, 2001*; *Herzberg et al., 2006*; *Laganenka et al., 2018*; *McLean et al., 1997*; *Paul et al., 2009*; *González Barrios et al., 2006*; *Yarwood et al., 2004*; *Koutsoudis et al., 2006*; *Waters and Bassler, 2005*; *Parsek and Greenberg, 2005*; *Jayaraman and Wood, 2008*; *Hentzer et al., 2003*; *Kirisits and Parsek, 2006*), we assume that planktonic cells transition to the biofilm state at a rate $\tau^{-1}$ when the local autoinducer concentration exceeds a threshold value $a^*$ (*Figure 1B*). Because our focus is on this transition, we assume that it is irreversible, and that cells in the biofilm lose motility. However, they still continue to consume nutrient, proliferate, and produce autoinducer with maximal rates $\kappa_2$, $\gamma_2$, and $k_2$ per cell, respectively; additional behaviors such as subsequent production of extracellular polymeric substances or transitioning back to the planktonic state can be incorporated as future extensions to this model.

Hence, while planktonic cells can disperse via active diffusion and chemotaxis, their dispersal is hindered—and biofilm formation is instead promoted—when autoinducer accumulates sufficiently, as schematized in *Figure 1A–B*. The central goal of this paper is to examine the processes underlying this competition between dispersal and biofilm formation. Our model is thus summarized as:

$$\text{Planktonic}: \frac{\partial b_1}{\partial t} = \underbrace{D_1 \nabla^2 b_1 - \nabla \cdot (b_1 \vec{v}_c)}_{\text{Motility}} + \underbrace{b_1 \gamma_1 g(c)}_{\text{Proliferation}}$$
$$- \underbrace{b_1 \tau^{-1} \mathcal{H}\left(a - a^*\right)}_{\text{Biofilm formation}} \tag{1}$$

$$\text{Biofilm}: \frac{\partial b_2}{\partial t} = \underbrace{b_2 \gamma_2 g(c)}_{\text{Proliferation}} + \underbrace{b_1 \tau^{-1} \mathcal{H}\left(a - a^*\right)}_{\text{Biofilm formation}} \tag{2}$$

$$\text{Nutrient}: \frac{\partial c}{\partial t} = \underbrace{D_c \nabla^2 c}_{\text{Diffusion}} - \underbrace{(b_1 \kappa_1 + b_2 \kappa_2) \, g(c)}_{\text{Consumption}} \tag{3}$$

$$\text{Autoinducer}: \frac{\partial a}{\partial t} = \underbrace{D_a \nabla^2 a}_{\text{Diffusion}} + \underbrace{(b_1 k_1 + b_2 k_2) \, g(c)}_{\text{Production}}$$
$$- \underbrace{a \left( \lambda + \alpha b_1 \right)}_{\text{Loss}} \tag{4}$$

where $\mathcal{H}$ is the Heaviside step function describing the transition from the planktonic to biofilm state. To explore the competition between motility-mediated dispersal and autoinducer-mediated biofilm formation, we then numerically solve this system of coupled equations using values of all parameters—which are either intrinsic descriptors of cellular physiology or are solely/additionally influenced by the local environment—that are derived from experiments (*Supplementary file 1*). Further details are provided in the Materials and methods. Additional simulations indicate that the results obtained are not appreciably influenced by variations in the exact nature of how our model treats the arrest in planktonic cell motility while transitioning to the biofilm state (*Figure 1—figure supplement 1*) or the initial inoculum shape (*Figure 1—figure supplement 2*).

### Representative numerical simulations

The results of a prototypical example are shown in *Figure 1C* and *Video 1*. Consumption by the planktonic cells (green curves) rapidly establishes a steep nutrient gradient (purple) at the leading edge of the inoculum. This gradient forces the planktonic cells to then move outward via chemotaxis.

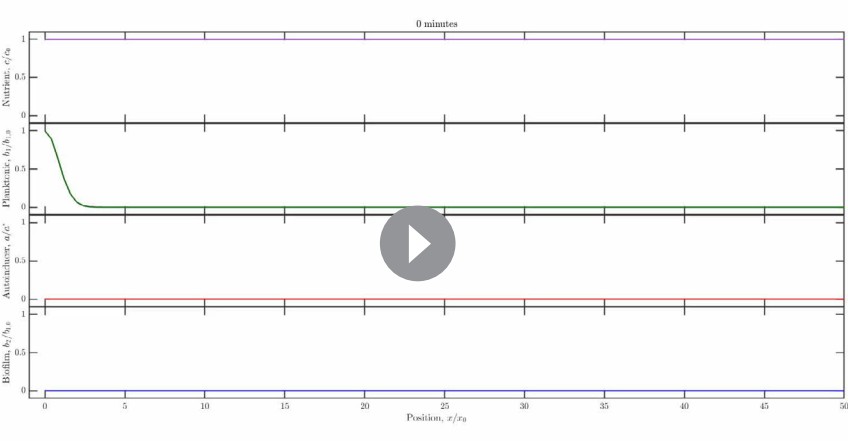

**Video 1.** Animated form of Figure 1C: Results of an example simulation of *Equations 1–4* showing the dynamics of the nutrient, planktonic cells, autoinducer, and biofilm cells from top to bottom, quantified by the normalized concentrations $c/c_0$, $b_1/b_{1,0}$, $a/a^*$, and $b_2/b_{1,0}$, respectively; $c_0$, $b_{1,0}$, and $a^*$ represent the initial nutrient concentration, initial bacterial concentration, and autoinducer threshold for biofilm formation, respectively. The position coordinate is represented by the normalized position $x/x_0$, where $x_0$ is the width of the initial cellular inoculum. The inoculum initially centered about the origin consumes nutrient (purple), establishing a gradient that drives outward dispersal by chemotaxis (outward moving green curves); the cells also produce autoinducer (red) concomitantly. At $t \approx 13$ h, sufficient autoinducer has been produced to trigger biofilm formation at the origin; at even longer times ($t \gtrsim 16$ h), nutrient depletion limits autoinducer production at this position. However, accumulation of autoinducer by the dispersing planktonic cells triggers partial biofilm formation at $x/x_0 \approx 4$ as well. This competition between dispersal and biofilm formation leads to a final biofilm fraction of $f = 21\%$ at the final time of $t = 20$ h. The values of the simulation parameters are given in *Supplementary file 2*. The video displays the profiles every 30 min, to retain a manageable file size; however, the temporal step size in the actual simulations is 0.1 s.

https://elifesciences.org/articles/76380/figures#video1

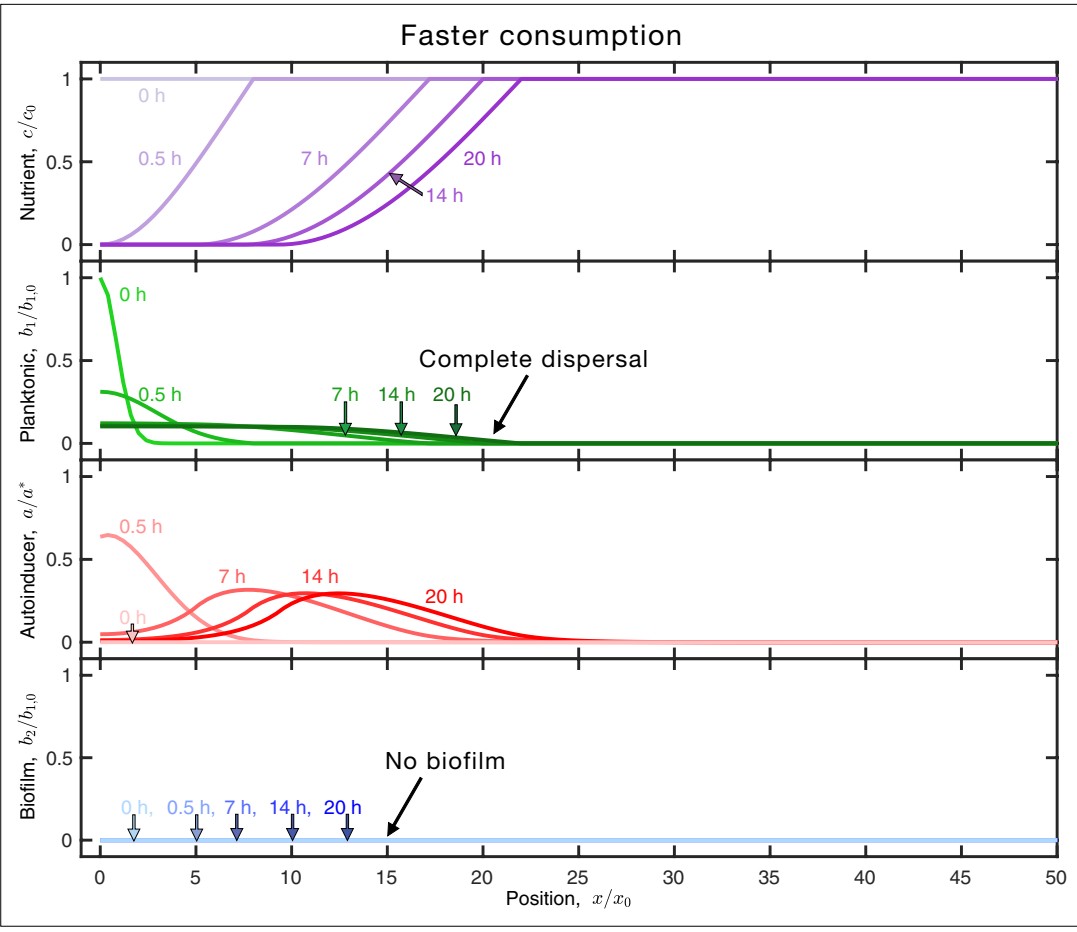

**Figure 2.** Faster nutrient consumption limits autoinducer production, leading to complete dispersal. Retion as in *Figure 1C*, but for planktonic cells with faster nutrient consumption (larger $\kappa_1$). Panels and colors show the same quantities as in *Figure 1C*. The inoculum initially centered about the origin consumes nutrient (purple), establishing a gradient that drives outward dispersal by chemotaxis (outward moving green curves); the cells also produce autoinducer (red) concomitantly. However, nutrient is depleted at this position more rapidly, limiting autoinducer production; as a result, the population continues to disperse in the planktonic state and the final biofilm fraction is $f = 0\%$. An animated form of this figure is shown in *Video 2*. The values of the simulation parameters are given in *Supplementary file 2*.

The online version of this article includes the following source data and figure supplement(s) for figure 2:

**Source data 1.**

**Figure supplement 1.** Slower nutrient consumption allows greater autoinducer production, leading to more biofilm formation.

**Figure supplement 1—source data 1.**

---

In particular, they self-organize into a coherent front that expands from the initial inoculum and continually propagates, sustained by continued consumption of the surrounding nutrient—consistent with the findings of previous studies of planktonic bacteria (*Bhattacharjee et al., 2021*). In this case, however, the cells also concomitantly produce autoinducer that accumulates into a growing plume (red). In some locations, the autoinducer eventually exceeds the threshold $a^*$, thus driving the formation of an immobilized biofilm (blue). Hence, at long times, $f = 21\%$ of the overall population is biofilm-associated, while the remaining $1 - f = 79\%$ continues to disperse in the planktonic state.

Because the processes underlying motility-mediated dispersal and autoinducer-mediated biofilm formation are highly species- and environment-dependent, the values of the parameters in *Equations 1–4* can span broad ranges—giving rise to different emergent behaviors under different conditions. Our simulations provide a way to examine how these behaviors depend on cellular concentration

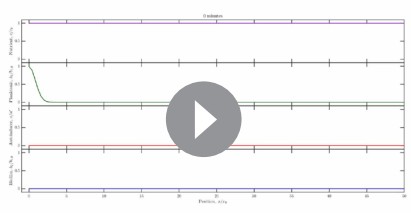

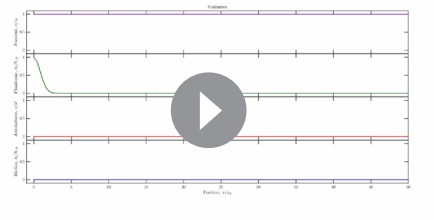

**Video 2.** Animated form of Figure 2: Results of the same simulation as in *Video 1*, but for planktonic cells with faster nutrient consumption (larger $\kappa_1$). Panels and colors show the same quantities as in *Video 1*. The inoculum initially centered about the origin consumes nutrient (purple), establishing a gradient that drives outward dispersal by chemotaxis (outward moving green curves); the cells also produce autoinducer (red) concomitantly. However, nutrient is depleted at this position more rapidly, limiting autoinducer production; as a result, the population continues to disperse in the planktonic state and the final biofilm fraction is $f = 0\%$. The values of the simulation parameters are given in *Supplementary file 2*. The video displays the profiles every 30 min, to retain a manageable file size; however, the temporal step size in the actual simulations is 0.1 s. https://elifesciences.org/articles/76380/figures#video2

**Video 3.** Animated form of Figure 2—figure supplement 1: Results of the same simulation as in *Video 1*, but for planktonic cells with slower nutrient consumption (smaller $\kappa_1$). Panels and colors show the same quantities as in *Video 1*. The inoculum initially centered about the origin slowly consumes nutrient (purple), establishing a slight gradient that allows partial planktonic dispersal (green curves moving outward); the cells also produce autoinducer (red) concomitantly. Because nutrient is consumed slowly, autoinducer production is not limited, resulting in partial biofilm formation (blue). Autoinducer has sufficiently accumulated above the threshold after $t \approx 14$ h, which causes a population of biofilm cells to form at the origin ($x/x_0 \approx 0$). After 20 h, the biofilm population continues to grow, and additionally, autoinducer concentration exceeds the threshold concentration at $x/x_0 \approx 10$. Thus, we see a second population of biofilm cells form, centered at $x/x_0 \approx 10$. The slower nutrient consumption results in a greater final biofilm fraction than in *Video 1*—here, $f = 52\%$. The values of the simulation parameters are given in *Supplementary file 2*. The video displays the profiles every 30 min, to retain a manageable file size; however, the temporal step size in the actual simulations is 0.1 s. https://elifesciences.org/articles/76380/figures#video3

and motility, quantified by $\{b_{1,0}, D_1, \chi_1, c_-, c_+\}$, nutrient availability and consumption, quantified by $\{D_c, c_0, \kappa_1, \kappa_2, c_{\text{char}}\}$, cellular proliferation, quantified by $\{\gamma_1, \gamma_2\}$, and autoinducer production, availability, and sensing, quantified by $\{D_a, k_1, k_2, \lambda, \alpha, \tau, a^*\}$. For example, implementing the same simulation as in *Figure 1C*, but for cells with faster nutrient consumption, yields a population that completely disperses in the planktonic state (the fraction of the population in the biofilm state at the final time of $t = 20$ h is $f = 0\%$, as shown in *Figure 2* and *Video 2*). Conversely, when cells consume nutrient slower, a larger fraction of the population forms an immobilized biofilm ($f = 52\%$, *Figure 2—figure supplement 1* and *Video 3*).

Given that the competition between motility-mediated dispersal and autoinducer-mediated biofilm formation depends sensitively on such a bewildering array of cellular and environmental factors, we ask whether these dependencies can be captured by simple, generalizable, biophysical rules. Nondimensionalization of *Equations 1–4* yields characteristic quantities and dimensionless groups that can parameterize these dependencies, as detailed in Appendix 1; however, given the large number of such groups, we seek an even simpler representation of the underlying processes that could unify the influence of all these different factors. To do so, we examine the fundamental processes underlying biofilm formation in our model.

## Availability of nutrient for autoinducer production

When autoinducer production is nutrient-dependent, we expect that a necessary condition for biofilm formation is that enough nutrient is available for sufficient autoinducer to be produced to eventually exceed the threshold $a^*$. To quantify this condition, we estimate two time scales: $\tau_d$, the time taken by the population of planktonic cells to deplete all the available nutrient locally, and $\tau_a$, the time at which produced autoinducer reaches the threshold for biofilm formation. While $\tau_d$ and $\tau_a$ can be directly obtained in each simulation, we seek a more generally-applicable analytical expression for both, solely using parameters that act as inputs to the model. In particular, for simplicity, we consider

nutrient consumption and autoinducer production, both occurring at their maximal rates $\kappa_1$ and $k_1$, respectively, by an exponentially-growing population of planktonic cells that are uniformly distributed in a well-mixed and fixed domain. Integrating *Equations 3 and 4* then yields (Appendix 2)

$$\tau_d = \gamma_1^{-1} \ln(1 + \tilde{\beta}_{1,0}) \tag{5}$$

$$\tau_a = \gamma_1^{-1} \ln \left[ 1 - \tilde{\zeta}_{1,0}^{-1} \ln \left( 1 - \tilde{\eta} \right) \right] . \tag{6}$$

Three key dimensionless quantities, denoted by the tilde (~) notation, emerge from this calculation. The first, $\tilde{\beta}_{1,0} \equiv \gamma_1 / \left( b_{1,0} \kappa_1 / c_0 \right)$, describes the yield of new cells produced as the population consumes nutrient—quantified by the rates of cellular proliferation and nutrient consumption, $\gamma_1$ and $b_{1,0}\kappa_1/c_0$, respectively (*Amchin et al., 2022*). The second, $\tilde{\eta} \equiv \alpha a^*/k_1$, describes the competition between auto-inducer loss and production, quantified by their respective rates $\alpha a^*$ and $k_1$, at the single-cell scale. The third, $\tilde{\zeta}_{1,0} \equiv \alpha b_0/\gamma_1$, describes the loss of autoinducer due to cell-surface binding as the population continues to grow, quantified by the population-scale rates of autoinducer loss and cellular proliferation, $\alpha b_0$ and $\gamma_1$, respectively; for simplicity, this quantity neglects natural degradation of autoinducer, given that the degradation rate is relatively small, with $\lambda \ll \alpha b_0$.

The ratio between *Equations 5 and 6* then defines a *nutrient availability parameter*, $\tilde{\mathcal{D}} \equiv \tau_d/\tau_a$. When $\tilde{\mathcal{D}}$ is large, produced autoinducer rapidly reaches the threshold for biofilm formation before the available nutrient is depleted; by contrast, when $\tilde{\mathcal{D}}$ is small, nutrient depletion limits autoinducer production. Hence, we hypothesize that $\tilde{\mathcal{D}} \gtrsim \tilde{\mathcal{D}}^*$ specifies a necessary condition for biofilm formation, where $\tilde{\mathcal{D}}^*$ is a threshold value of order unity. The simulations shown in *Figures 1C and 2* and *Figure 2—figure supplement 1* enable us to directly test this hypothesis. Consistent with our expectation, the simulation in *Figure 1C* is characterized by $\tilde{\mathcal{D}} = 0.33$, near the expected threshold for biofilm formation; as a result, $f = 21\%$. When consumption is faster as in *Figure 2* ($\tilde{\mathcal{D}} = 0.033$), the available nutrient is rapidly depleted; thus, cells disperse away before sufficient autoinducer is produced to initiate biofilm formation, and $f = 0\%$. Conversely, when nutrient consumption is slow as in *Figure 2—figure supplement 1* ($\tilde{\mathcal{D}} = 3.1$), nutrient continues to be available for autoinducer production, eventually driving biofilm formation, with a larger fraction $f = 52\%$.

Taken together, these results support our hypothesis that $\tilde{\mathcal{D}} \gtrsim \tilde{\mathcal{D}}^* \sim 1$ is a necessary condition for biofilm formation. It is not, however, a sufficient condition: repeating the simulation of *Figure 1C* but for faster-moving cells yields a population that rapidly disperses without forming a biofilm at all ($f = 0\%$, *Figure 3A* and *Video 4*)—despite having the same value of $\tilde{\mathcal{D}} = 0.33$. Thus, our mathematical description of the conditions that determine biofilm formation is, as yet, incomplete.

## Competition between motility-mediated dispersal and autoinducer accumulation

The results shown in *Figure 3* indicate that the ability of planktonic bacteria to move, which is not incorporated in the nutrient consumption parameter $\tilde{\mathcal{D}}$, also plays a key role in regulating whether a biofilm forms. Indeed, close inspection of *Figure 3A* hints at another necessary condition for biofilm formation: as shown by the magnified view in *Figure 3B* (e.g., at $t = 4\,\mathrm{h}$), the leading edge of the dispersing planktonic cells extends beyond the plume of produced autoinducer. Therefore, we expect that even when sufficient nutrient is available for autoinducer production ($\tilde{\mathcal{D}} \gtrsim \tilde{\mathcal{D}}^* \sim 1$), autoinducer production must be rapid enough to reach the threshold for biofilm formation before cells have dispersed away. To quantify this condition, we estimate the the time $\tau_c$ at which the motile plank-tonic cells begin to 'outrun' the growing autoinducer plume. Specifically, we quantify the dynamics of the leading edge positions of the chemotactic front of planktonic cells and the autoinducer plume, $x_{1,\mathrm{edge}}(t)$ and $x_{a,\mathrm{edge}}(t)$, respectively. The front position $x_{1,\mathrm{edge}}(t)$ is known to depend on cellular motility, nutrient diffusion, and nutrient consumption in a non-trivial manner (*Berg, 2004*; *Cremer et al., 2019*; *Fu et al., 2018*; *Amchin et al., 2022*), and we are not aware of a way to compute this quantity a priori from input parameters; instead, we extract this sole quantity from each simulation by identifying the largest value of $x$ at which $b_1 \geq 10^{-4}b_{1,0}$. While the plume position $x_{a,\mathrm{edge}}(t)$ can also be directly obtained in each simulation, we again develop a more generally applicable analytical expression by assuming that the autoinducer continually diffuses from the initial inoculum: $x_{a,\mathrm{edge}}(t) = x_0 + \sqrt{2D_a t}$. Then, $\tau_c$ can be directly determined as the time at which $x_{1,\mathrm{edge}}(t)$ begins to exceed $x_{a,\mathrm{edge}}(t)$.

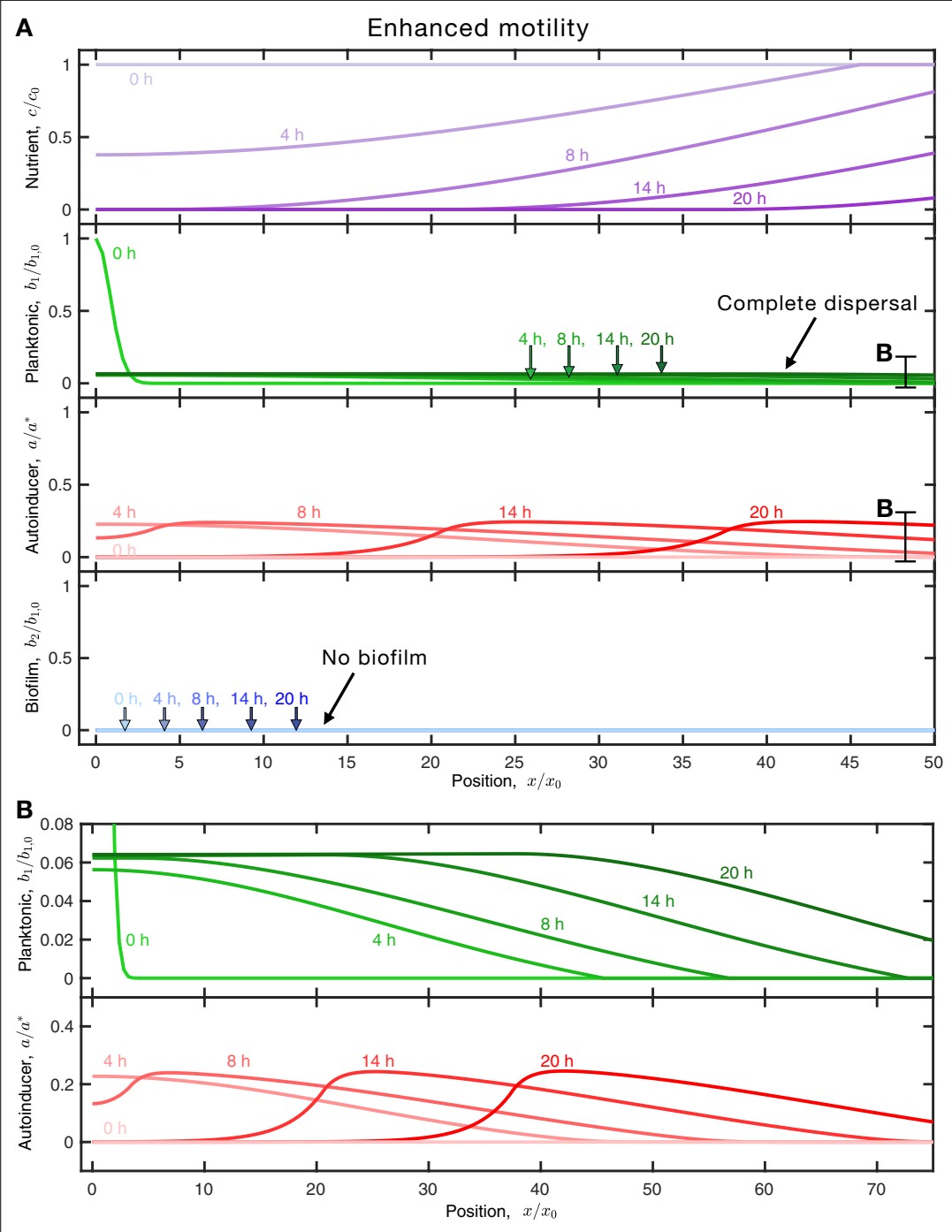

**Figure 3.** Enhanced motility enables cells to disperse before sufficient autoinducer accumulates, leading to complete dispersal. (**A**) Results of the same simulation as in *Figure 1C*, but for faster-moving planktonic cells (larger $D_1$ and $\chi_1$). Panels and colors show the same quantities as in *Figure 1C*. The inoculum initially centered about the origin consumes nutrient (purple), establishing a gradient that drives outward dispersal by chemotaxis (outward moving green curves); the cells also produce autoinducer (red) concomitantly. More rapid dispersal enables the planktonic cells to 'outrun' the growing autoinducer plume, as shown by the extended and magnified view in (**B**). As a result, the population continues to disperse in the planktonic state and the final biofilm fraction is $f = 0\%$. An animated form of this figure is shown in *Video 4*. The values of the simulation parameters are given in *Supplementary file 2*.

The online version of this article includes the following source data and figure supplement(s) for figure 3:

*Figure 3 continued on next page*

*Figure 3 continued*

**Source data 1.**

**Figure supplement 1.** Diminished motility enables autoinducer to accumulate, resulting in increased biofilm formation.

**Figure supplement 1—source data 1.**

The ratio between $\tau_c$ thereby determined and $\tau_a$, the time required for produced autoinducer to reach the threshold for biofilm formation (*Equation 6*), then defines a *cellular dispersal parameter*, $\tilde{\mathcal{J}} \equiv \tau_c/\tau_a$. When $\tilde{\mathcal{J}}$ is large, autoinducer accumulation is sufficiently rapid to drive biofilm formation; by contrast, when $\tilde{\mathcal{J}}$ is small, the planktonic cells rapidly disperse without forming a biofilm. Hence, we hypothesize that $\tilde{\mathcal{J}} \gtrsim \tilde{\mathcal{J}}^*$ specifies another necessary condition for biofilm formation, where $\tilde{\mathcal{J}}^*$ is, again, a threshold value of order unity. The simulations shown in *Figures 1C and 3A* enable us to directly test this hypothesis. Consistent with our expectation, the simulations in *Figure 1C* and *Figure 2—figure supplement 1* are characterized by $\tilde{\mathcal{J}} = 1.6$, near the expected threshold for biofilm formation; as a result, $f > 0$ in both cases. Furthermore, implementing the same simulation as *Figure 1C* (with the same $\tilde{\mathcal{D}} = 0.33$) but for slower-moving cells, characterized by a larger $\tilde{\mathcal{J}} = 120$, yields a population that forms an even larger biofilm fraction $f = 82\%$ (*Figure 3—figure supplement 1* and *Video 5*). Conversely, when cellular dispersal is faster as in *Figure 3*, characterized by a smaller $\tilde{\mathcal{J}} = 0.1$, the cells disperse away before sufficient autoinducer is produced to initiate biofilm formation, and $f = 0\%$. Taken together, these results support our hypothesis that $\tilde{\mathcal{J}} \gtrsim \tilde{\mathcal{J}}^* \sim 1$ is another necessary condition for biofilm formation.

## A universal biophysical threshold for biofilm formation

Thus far, we have shown that the two conditions $\tilde{\mathcal{D}} \gtrsim \tilde{\mathcal{D}}^*$ and $\tilde{\mathcal{J}} \gtrsim \tilde{\mathcal{J}}^*$ are both necessary for biofilm formation. Is the combination of both sufficient to fully specify the conditions required for biofilm formation? To test this possibility, we implement

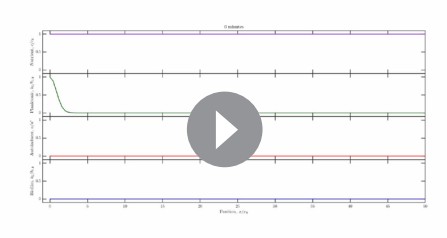

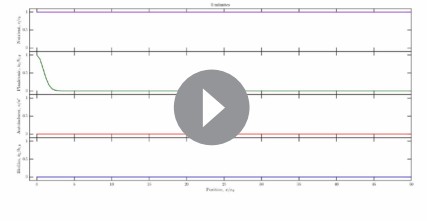

**Video 4.** Animated form of Figure 3: Results of the same simulation as in *Video 1*, but for faster-moving planktonic cells (larger $D_1$ and $\chi_1$). Panels and colors show the same quantities as in *Video 1*. The inoculum initially centered about the origin consumes nutrient (purple), establishing a gradient that drives outward dispersal by chemotaxis (outward moving green curves); the cells also produce autoinducer (red) concomitantly. More rapid dispersal enables the planktonic cells to 'outrun' the growing autoinducer plume, as shown by the extended and magnified view in (B). As a result, the population continues to disperse in the planktonic state and the final biofilm fraction is $f = 0\%$. The values of the simulation parameters are given in *Supplementary file 2*. The video displays the profiles every 30 min, to retain a manageable file size; however, the temporal step size in the actual simulations is 0.1 s.

https://elifesciences.org/articles/76380/figures#video4

**Video 5.** Animated form of Figure 3—figure supplement 1: Results of the same simulation as in *Video 1*, but for slower-moving planktonic cells (smaller $D_1$ and $\chi_1$). Panels and colors show the same quantities as in *Video 1*. The inoculum initially centered about the origin consumes nutrient (purple), establishing a slight gradient—however, because the motility parameters are diminished, the planktonic population (green) remains around the origin. The planktonic cells produce autoinducer (red) concomitantly, and after 1 h, the autoinducer concentration exceeds the threshold concentration. Thus, some of the planktonic cells transition to biofilm cells, centered at the origin. Both the biofilm cells and planktonic cells continue to grow, produce autoinducer, and consume nutrient; the planktonic cells do not disperse due to their diminished motility, resulting in a larger fraction of biofilm cells ($f = 82\%$) than in *Video 1*. The values of the simulation parameters are given in . The video displays the profiles every 30 min, to retain a manageable file size; however, the temporal step size in the actual simulations is 0.1 s.

https://elifesciences.org/articles/76380/figures#video5

10,983 numerical simulations of *Equations 1–4* exploring the full physiological ranges of the input parameters that describe cellular, nutrient, and autoinducer properties for diverse bacterial species/strains and environmental conditions (*Supplementary file 1*). For each simulation, we compute $\tilde{\mathcal{D}}$, $\tilde{\mathcal{J}}$, and $f$. Remarkably, despite the extensive variability in the values of the underlying parameters, all the results cluster between two states parameterized by $\tilde{\mathcal{D}}$ and $\tilde{\mathcal{J}}$, as shown in *Figure 4A*: motility-mediated dispersal without biofilm formation ($f = 0\%$, green points) when either $\tilde{\mathcal{D}} < \tilde{\mathcal{D}}^*$ or $\tilde{\mathcal{J}} < \tilde{\mathcal{J}}^*$, and biofilm formation without dispersal ($f = 100\%$, blue points) when both $\tilde{\mathcal{D}} > \tilde{\mathcal{D}}^*$ and $\tilde{\mathcal{J}} > \tilde{\mathcal{J}}^*$. Many different combinations of the input parameters yield the same $(\tilde{\mathcal{D}}, \tilde{\mathcal{J}})$; yet, no matter the input values of these parameters, which vary over broad ranges for different cells and environmental conditions, $(\tilde{\mathcal{D}}, \tilde{\mathcal{J}})$ uniquely specify the resulting biofilm fraction $f$ for all points, as shown in *Figure 4B–C*—indicating that these two dimensionless parameters reasonably encompass all the factors determining biofilm formation within our model. We observe some exceptions at the boundary between these two states, likely because the simplifying assumptions underlying the derivation of the $\tilde{\mathcal{D}}$ and $\tilde{\mathcal{J}}$ parameters begin to break down. Nevertheless, the boundary between both states, summarized by the relation $\tilde{\mathcal{D}}^*/\tilde{\mathcal{D}} + \tilde{\mathcal{J}}^*/\tilde{\mathcal{J}} \sim 1$ with $\tilde{\mathcal{D}}^*$ and $\tilde{\mathcal{J}}^*$ both $\sim 1$ (black curve), thus specifies a universal biophysical threshold for biofilm formation.

## Discussion

The transition from the planktonic to biofilm state is known to depend on a large array of factors that describe cellular concentration, motility, and proliferation; nutrient availability and consumption; and autoinducer production, availability, and sensing—all of which can vary considerably for different strains/species of bacteria and environmental conditions. Therefore, quantitative prediction of the onset of biofilm formation is challenging. The biophysical model presented here provides a key step toward addressing this challenge. In particular, for the illustrative case we consider—in which cells can either disperse through active motility, retaining them in the planktonic state, or form an immobilized biofilm when exposed to sufficient autoinducer—we have shown that the onset of biofilm formation is uniquely specified by a biophysical threshold set by the two dimensionless parameters $\tilde{\mathcal{D}}$ (quantifying nutrient availability) and $\tilde{\mathcal{J}}$ (quantifying bacterial dispersal). Importantly, within the formulation of our model, this threshold is universal: many different combinations of cellular and environmental factors are described by the same $(\tilde{\mathcal{D}}, \tilde{\mathcal{J}})$, and thus, yield the same onset of biofilm formation. Therefore, given a bacterial strain and set of environmental conditions, extensions of our model could help provide a way to predict whether a biofilm will form a priori. Indeed, because the factors that define $\tilde{\mathcal{D}}$ and $\tilde{\mathcal{J}}$ can be directly measured, our work now provides quantitative principles and predictions (as summarized in *Figure 4*) to guide future experiments.

For generality, our model also incorporates proliferation, nutrient consumption, and autoinducer production by cells after they have transitioned to the biofilm state. Hence, within our model, biofilm-produced autoinducer could also drive surrounding planktonic cells to transition to the biofilm state. In this case, we expect that the long-time fraction of the population in the biofilm state, $f$, will also depend on nutrient depletion and autoinducer production by the growing biofilm. Indeed, performing a similar calculation as that underlying the nutrient availability parameter, $\tilde{\mathcal{D}}$, yields a third dimensionless parameter, $\tilde{\mathcal{S}} \equiv \tau_{d,2}/\tau_{a,2}$; here, $\tau_{d,2}$ and $\tau_{a,2}$ describe the times at which biofilm cells have depleted all the available nutrient and produced enough autoinducer to reach the threshold for biofilm formation, respectively (Appendix 2). Thus, we hypothesize that, while the *onset* of biofilm formation is specified by $(\tilde{\mathcal{D}}, \tilde{\mathcal{J}})$, the final *extent* of biofilm that has formed will also be described by $\tilde{\mathcal{S}}$. The results shown in *Figures 1–4C* have a fixed $\tilde{\mathcal{S}} = 50$, which describes the case of a biofilm that produces autoinducer rapidly; repeating these simulations for the opposite case of slow autoinducer production by biofilm cells, with $\tilde{\mathcal{S}} = 1/50$, yields the state diagram shown in *Figure 4D*. In agreement with our hypothesis, while the transition to the biofilm state (black line) is not appreciably altered by the change in $\tilde{\mathcal{S}}$, the transition to complete biofilm formation ($f = 1$) is more gradual in this case (compare *Figure 4A,B,D,E*). Moreover, we note that our analysis thus far has focused on the case in which autoinducer production is nutrient-dependent; however, this process may sometimes be nutrient-independent (*Narla et al., 2021*). In this case, we expect that our overall analysis still applies, but with the onset of biofilm formation specified by only the dispersal parameter $\tilde{\mathcal{J}}$—as confirmed in *Figure 4—figure supplement 1*.

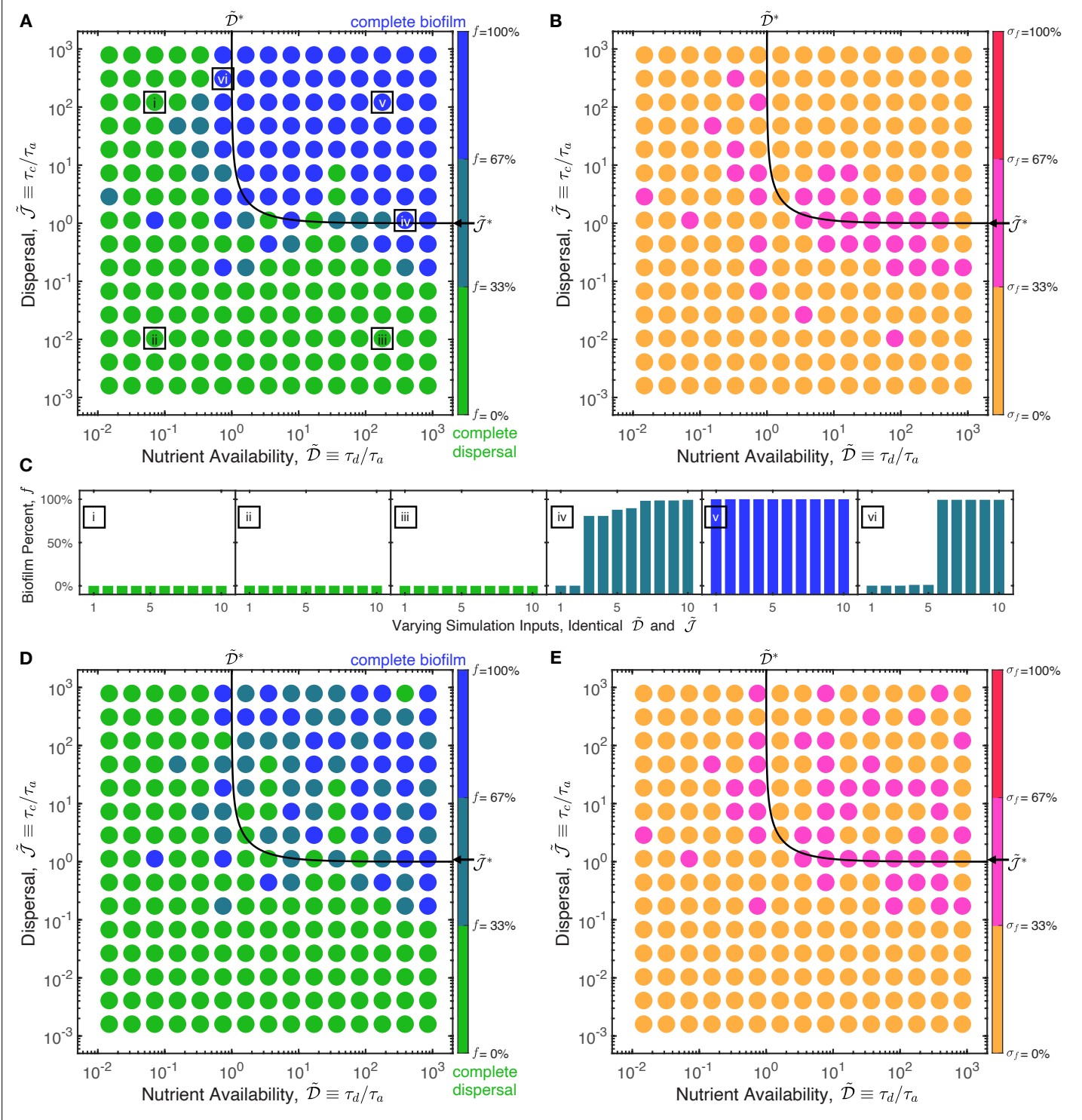

**Figure 4.** The two states of complete dispersal by planktonic cells (green) and complete formation of a biofilm (blue) can be universally described by three dimensionless parameters. (**A**) State diagram showing the fraction of biofilm formed, $f$, at the final time ($t = 20$ h) for different values of the nutrient availability and cellular dispersal parameters, $\tilde{\mathcal{D}}$ and $\tilde{\mathcal{J}}$, respectively. The state diagram summarizes the results of 10,983 simulations of *Equations 1–4* exploring the full range of parameter values describing different bacterial species/strains and different environmental conditions (*Supplementary file 1*). Each point represents the mean value of $f$ obtained from multiple simulations with different parameter values, but with similar $\tilde{\mathcal{D}}$ and $\tilde{\mathcal{J}}$ (identical within each bin defined by the spacing between points). (**B**) represents the same data, but each point represents the standard deviation of the values of $f$ obtained from the same simulations. Despite the vastly differing conditions explored in each simulation, they cluster into the two states of planktonic dispersal (green) and biofilm formation (blue) when parameterized by $\tilde{\mathcal{D}}$ and $\tilde{\mathcal{J}}$. The boundary between the two states can

*Figure 4 continued on next page*

*Figure 4 continued*

be described by the relation $\tilde{\mathcal{D}}^*/\tilde{\mathcal{D}} + \tilde{\mathcal{J}}^*/\tilde{\mathcal{J}} \sim 1$, as shown by the black line; this relation combines the transition between the two states that occurs at both $\tilde{\mathcal{D}}^* \sim 1$ and $\tilde{\mathcal{J}}^* \sim 1$. Away from this boundary, all simulations for the same $\tilde{\mathcal{D}}$ and $\tilde{\mathcal{J}}$ collapse to have the same biofilm fraction $f$, as shown by the points in (**B**) and examples (i)–(iii) and (v) in (**C**)—confirming the universality of our parameterization. Near the boundary, we observe some slight differences between simulations, as shown in (**B**) and examples (iv) and (vi) in (**C**). The values of the simulation parameters for the examples in (**C**) are given in Source Data file 1. The data in (**A**–**C**) correspond to a fixed value of the third dimensionless parameter $\tilde{\mathcal{S}} = 50$, which describes the case of biofilm cells that produce autoinducer rapidly; repeating these simulations for the opposite case of slow autoinducer production by biofilm cells ($\tilde{\mathcal{S}} = 1/50$) yields the state diagram shown in (**D**), but for 14,351 simulation runs; again, (**E**) shows the standard deviation of the corresponding values of $f$. As shown by (**D**–**E**), while the transition between the two states (black line) is unaffected by the change in $\tilde{\mathcal{S}}$, the transition to complete biofilm formation is more gradual. Together, the three parameters $\tilde{\mathcal{D}}$, $\tilde{\mathcal{J}}$, and $\tilde{\mathcal{S}}$ provide a full description of the onset and extent of biofilm formation across vastly different conditions.

The online version of this article includes the following source data and figure supplement(s) for figure 4:

**Source data 1.**

**Source data 2.**

**Source data 3.**

**Figure supplement 1.** In the case of 'protected' nutrient-independent autoinducer production, the transition from planktonic to biofilm states occurs at, $\tilde{\mathcal{J}} \sim 1$ independent of $\tilde{\mathcal{D}}$.

**Figure supplement 1—source data 1.**

**Figure supplement 2.** Simulation results are not appreciably influenced by our choice of discretization.

**Figure supplement 2—source data 1.**

## Possible extensions of our work

The transition from the planktonic to biofilm state is highly complex and, in many cases, has features that are unique to different species of bacteria. Nevertheless, our model provides a minimal description that can capture many of the essential features of biofilm formation more generally—thereby providing a foundation for future extensions of our work, some of which are described below.

1. For simplicity, our model considers only one spatial dimension; however, fascinating new effects may arise in higher-dimensional implementations of our model. For example, in our prior work modeling the collective migration of planktonic bacteria in the *absence* of quorum sensing-mediated biofilm formation, we found that variations in the shape of the cellular front orthogonal to the main propagation direction 'smooth out' over time (*Alert et al., 2022*; *Bhattacharjee et al., 2022*). In particular, cells at outward-bulging parts of the front are exposed to more nutrient, which diminishes their ability to respond to the nutrient gradient via chemotaxis and thus slows them down. As a result, the migrating front eventually smooths to a flat shape whose subsequent dynamics can then be described using just one spatial dimension, just as in our treatment here. However, we expect that this behavior could be altered in interesting new ways when the cells can additionally produce and sense autoinducer and thereby transition to the biofilm state, as is the case here. In this case, we speculate that because cells at outward-bulging parts of the front are exposed to more nutrients and have a weaker chemotactic response, autoinducer production and accumulation will be more rapid relative to cellular dispersal. That is, at these parts of the front, $\tau_a$ and $\tau_c$ will be shorter and longer, respectively, causing the dispersal parameter $\tilde{\mathcal{J}}$ to be larger locally. Thus, our model would predict biofilm formation to occur first at these parts of the front, potentially also influencing subsequent dispersal and biofilm formation at other locations along the front. Therefore, while our conclusions here could be the same *locally* at different parts of the front, the *global* behavior of the population could be different—potentially giving rise to e.g., spatially-heterogeneous biofilm formation.

2. As an illustrative example, our model considers the case in which cells produce a single autoinducer; however, some quorum sensing systems utilize multiple autoinducers (*Miller and Bassler, 2001*; *Miller et al., 2002*; *Pesci et al., 1997*), which could be described using additional field variables and equations similar to *Equation 4*. Moreover, while we take the nutrient to be the sole chemoattractant, in some cases, autoinducers can also act as chemoattractants (*Laganenka et al., 2016*), which could also be described in our framework by e.g., introducing autoinducer-dependent chemotaxis in the drift velocity in *Equation 1*.

3. Our model considers positive quorum sensing control in which planktonic cells transition to the biofilm state in a step-like fashion when the local autoinducer concentration exceeds a threshold

value. That is, when planktonic cells encounter sufficiently concentrated autoinducer, the diffusivity and chemotactic coefficient transition in a step-like fashion from the constant values $D_1$ and $\chi_1$, respectively, to zero after the time duration $\tau$, for simplicity. In real systems, the change in cellular motility may not be as temporally abrupt. Future work could address a more gradual loss of motility in our theoretical framework by, for example, considering a cellular diffusivity and chemotactic coefficient that gradually transition from their planktonic values to zero over a non-zero time scale. Given that the same cells would be transitioning from the motile planktonic to immotile biofilm state—but in this case with the introduction of a time-varying diffusivity and chemotactic coefficient—we expect that the long-time biofilm fraction $f$ will be similar, and only the spatial profile of the biofilm population may be altered. Hence, we expect that our main findings summarized in *Figure 4* will be unaffected by such a change. Indeed, performing the same representative simulation shown in *Figure 1C*, but with both motility parameters $D_1$ and $\chi_1$ smoothly transitioning to zero in time, shows nearly identical results (*Figure 1—figure supplement 1*)—confirming our expectation that the temporal nature of the arrest in motility does not appreciably influence our model results and conclusions.

4. While we take the transition to the biofilm state as being irreversible, this is often not the case (*Barraud et al., 2006*; *Kaplan, 2010*; *Abdel-Aziz, 2014*). Longer-time transitions back to the planktonic state could be described using additional terms similar to the last terms of *Equations 1; 2*, but with the opposite sign. Similar modifications could be made to describe other species of bacteria (e.g., *Vibrio cholerae*) that utilize the opposite case of negative quorum sensing control, in which biofilm cells instead transition to the planktonic state when the autoinducer accumulates above a threshold value (*Hammer and Bassler, 2003*; *Bridges and Bassler, 2019*).

5. Biofilms are often formed by multiple different microbial species, whereas our model describes biofilm formation by a single species, for simplicity. Nevertheless, we expect that our theoretical framework can be extended by following reasoning similar to that described in this paper, but with the introduction of additional equations and variables in the governing *Equations 1–4* to describe the distinct cell and chemical types, as appropriate. For example, if the different species $i$ consume and respond to distinct nutrients $c_i$, and secrete and respond to distinct autoinducers $a_i$, each species could be described in isolation using our same governing *Equations 1–4*, but now extended to incorporate the distinct variables $c_i$, $a_i$, $b_{1,i}$, and $b_{2,i}$. Then, directly following our approach, each species would be described by its own dimensionless parameters $\tilde{\mathcal{D}}_i$ and $\tilde{\mathcal{J}}_i$, with $\tilde{\mathcal{D}}^*/\tilde{\mathcal{D}}_i + \tilde{\mathcal{J}}^*/\tilde{\mathcal{J}}_i \sim 1$ again specifying the threshold for biofilm formation for each. We hypothesize that the composition of the final two-species biofilm community would then be given by the combination of each single-species biofilm. Alternatively, in the case that the different species consume and respond to the same nutrient $c$, and secrete and respond to the same autoinducer $a$, our *Equations 1–4* could again be extended to consider the cellular parameters specific to each species $i$. In this approach, however, biofilm formation by each of the species cannot be described in isolation, because they are coupled through the nutrient and autoinducer dynamics. Instead, the calculations of the characteristic time scales $\tau_d$, $\tau_a$, and $\tau_c$ would need to be extended, following our approach, to now reflect contributions from all the different species. We hypothesize that the overall multi-species community would then be described by one set of governing dimensionless parameters $(\tilde{\mathcal{D}}, \tilde{\mathcal{J}})$, and $\tilde{\mathcal{D}}^*/\tilde{\mathcal{D}} + \tilde{\mathcal{J}}^*/\tilde{\mathcal{J}} \sim 1$ would again specify a universal biophysical threshold for the *onset* of biofilm formation for the overall community—but the *composition* of the final multi-species biofilm that results above this threshold may not be uniquely specified by $(\tilde{\mathcal{D}}, \tilde{\mathcal{J}})$.

6. Biofilm formation may be regulated by other, non-quorum sensing-based, processes not considered in our model. For example, the intracellular accumulation of secondary signaling molecules such as cyclic di-GMP can also regulate biofilm formation (*Valentini and Filloux, 2016*; *Simm et al., 2004*; *Jenal et al., 2017*; *Römling et al., 2013*; *Hengge, 2009*; *Krasteva et al., 2010*; *Baraquet and Harwood, 2013*; *Trampari et al., 2015*; *Davis et al., 2013*; *Boehm et al., 2010*; *Russell et al., 2013*). In some cases, this process may be controlled by quorum sensing (*Waters et al., 2008*) and thus could be described by our model, while in others, it is controlled by other cues such as e.g., contact with surfaces, which would need to additionally be incorporated into our theoretical framework.

7. Finally, we note that our model is deterministic—describing the cellular processes of motility, nutrient consumption, proliferation, and autoinducer production, availability, and sensing using the parameters $\{D_1, \chi_1, c_-, c_+, \kappa_1, \kappa_2, c_{\text{char}}, \gamma_1, \gamma_2, k_1, k_2, \alpha, \tau, a^*\}$, respectively. Each of these is taken to be single-valued in each of our simulations. However, these parameters can have a distribution of values arising from e.g., inherent cell-to-cell variability. Because these values define the governing $\tilde{\mathcal{D}}$ and $\tilde{\mathcal{J}}$ that specify the threshold for biofilm formation in our model, we

expect that variability in the parameter values would broaden the planktonic-to-biofilm transition predicted by our model. That is, we expect the transition specified by the black curve in *Figure 4A* to be smeared out, similar to what is seen in *Figure 4D*, though due to a fundamentally different reason—with biofilm formation arising in some cases at lower $(\tilde{\mathcal{D}}, \tilde{\mathcal{J}})$ than predicted by the black curve. Indeed, similar behavior was recently observed in a distinct model of biofilm formation on flat surfaces (*Sinclair et al., 2022*). Exploring the influence of such variations by using a more probabilistic approach in our theoretical framework will thus be a useful direction for future research.

## Materials and methods

To numerically solve the continuum model described by *Equations 1–4*, we follow the experimentally validated approach used in our previous work (*Bhattacharjee et al., 2021*; *Amchin et al., 2022*). Specifically, we use an Adams-Bashforth-Moulton predictor-corrector method in which the order of the predictor and corrector are 3 and 2, respectively. Because the predictor-corrector method requires past time points to inform future steps, the starting time points must be found with another method; we choose the Shanks starter of order 6 as described previously (*Rodabaugh and Wesson, 1965*; *Shanks, 1966*). For the first and second derivatives in space, we use finite difference equations with central difference forms in rectilinear coordinates. The temporal and spatial resolution of the simulations are $\delta t = 0.1$ s and $\delta x = 20$ $\mu$m, respectively; furthermore, we constrain our analysis to simulations for which the peak of the overall bacteria population moves slower than $\delta x / \delta t$. Repeating representative simulations with different spatial and temporal resolution indicates that even finer discretization does not appreciably alter the results (*Figure 4—figure supplement 2*). Thus, our choice of discretization is sufficiently finely-resolved such that the results in the numerical simulations are not appreciably influenced by discretization. Furthermore, performing the same representative simulation shown in *Figure 1C*, but with the shape of the initial inoculum changed from a Gaussian profile to a step function with the same maximum cellular concentration and width, shows nearly identical results (*Figure 1—figure supplement 2*)—suggesting that our results are robust to variations in this initial condition chosen. Further probing the mathematical structure of our biophysical model to examine additional influences of initial conditions and explore the possibility of oscillatory solutions, closed orbits, or singularities would be a fascinating direction for future work.

To connect the simulations to our previous experiments (*Bhattacharjee et al., 2021*), we choose a total extent of $1.75 \times 10^4$ $\mu$m for the size of the entire simulated system, with no-flux conditions for the field variables $b_1$, $b_2$, $c$, and $a$ applied to both boundaries at $x = 0$ and $1.75 \times 10^4$ $\mu$m. As in the experiments, we initialize each simulation with a starting inoculum of planktonic cells with a Gaussian profile defined by the maximum concentration $b_{1,0}$ at $x = 0$ and a full width at half maximum of $100$ $\mu$m. Nutrient is initially uniform at a fixed concentration $c_0$, and the autoinducer and biofilm concentrations are initially zero, throughout. Furthermore, following previous work (*Amchin et al., 2022*; *Dell'Arciprete et al., 2018*; *Volfson et al., 2008*; *Farrell et al., 2013*; *Klapper and Dockery, 2002*; *Head, 2013*), we also incorporate jammed growth expansion of the population in which growing cells push outward on their neighbors when the total concentration of bacteria is large enough. In particular, whenever the total concentration of bacteria (planktonic and biofilm) exceeds the jamming limit of $0.95$ cells $\mu$m$^{-3}$ at a location $x_i$, the excess cell concentration is removed from $x_i$ and added to the neighboring location, $x_i + \delta x$, where $\delta x$ represents the spatial resolution of the simulation, retaining the same ratio of planktonic to biofilm cells in the new location. We repeat this process for every location in the simulated space for each time step.

We run each simulation for a total simulated duration of $t_{\text{sim}} = 20$ h. At this final time, we use the simulation data to directly compute $f \equiv \frac{\int b_2 \mathrm{d}x}{\int b_2 \mathrm{d}x + \int b_1 \mathrm{d}x}$, the total fraction of the population in the biofilm state. We also compute the values of the dimensionless parameters $\tilde{\mathcal{D}}$, $\tilde{\mathcal{J}}$, and $\tilde{\mathcal{S}}$ using the equations presented in the main text. We note that the autoinducer production time $\tau_a$ (*Equation 6*) is only finite for $\tilde{\eta} \equiv \alpha a^* / k_1 < 1$; when $\tilde{\eta} \geq 1$, the rate of autoinducer loss exceeds that of autoinducer production, and thus the time required to reach the threshold for biofilm formation diverges. Because both $\tilde{\mathcal{D}}$ and $\tilde{\mathcal{J}}$ are defined as $\tau_d / \tau_a$ and $\tau_c / \tau_a$, respectively, for simulations with $\tilde{\eta} \geq 1$, we represent them on the state diagrams in *Figure 4* and *Figure 4—figure supplement 1* at $(\mathcal{D}, \mathcal{J}) = (10^{-2}, 10^{-3})$, the smallest values shown on the diagrams. All of these simulations have $f = 0$, as expected. Furthermore, to

ensure that $t_{\text{sim}}$ is sufficiently long, we (i) only perform simulations with $\tau_a$ and $\tau_{a,2}$ smaller than $t_{\text{sim}}$, and (ii) do not include simulations with $f = 0$ but $\tau_c = \tau_{\text{sim}}$, for which sufficient time has not elapsed for planktonic cells to chemotactically disperse.

## Acknowledgements

It is a pleasure to acknowledge R Kōnane Bay, Vernita Gordon, Anna M Hancock, Andrej Košmrlj, Alejandro Martínez-Calvo, JTN Moore-Ott, PG-A Moore-Ott, NA Moore-Ott, Howard Stone, Carolina Trenado-Yuste, and Ned Wingreen for stimulating discussions. This work was supported by NSF grants CBET-1941716 and EF-2124863, the Eric and Wendy Schmidt Transformative Technology Fund at Princeton, the Princeton Catalysis Initiative, a Reiner G Stoll Undergraduate Summer Fellowship (to SC), in part by funding from the Princeton Center for Complex Materials, a Materials Research Science and Engineering Center supported by NSF grant DMR-2011750, and the Pew Charitable Trusts through the Pew Biomedical Scholars Program. This material is also based upon work supported by the National Science Foundation Graduate Research Fellowship Program (to JAM-O) under Grant No. DGE-1656466. Any opinions, findings, and conclusions or recommendations expressed in this material are those of the authors and do not necessarily reflect the views of the National Science Foundation. This publication was supported by the Princeton University Library Open Access Fund.

## Additional information

### Funding

| Funder | Grant reference number | Author |
| --- | --- | --- |
| National Science Foundation | CBET-1941716 | Sujit S Datta |
| National Science Foundation | EF-2124863 | Sujit S Datta |
| National Science Foundation | DMR-2011750 | Sujit S Datta |
| Pew Charitable Trusts | Pew Biomedical Scholars Program | Sujit S Datta |
| National Science Foundation | DGE-1656466 | Jenna A Moore-Ott |
| Princeton University | Eric and Wendy Schmidt Transformative Technology Fund | Sujit S Datta |
| Princeton University | Princeton Catalysis Initiative | Sujit S Datta |
| Princeton University | Reiner G. Stoll Undergraduate Summer Fellowship | Selena Chiu |
| Princeton University | Princeton University Library Open Access Fund | Jenna A Moore-Ott |

The funders had no role in study design, data collection and interpretation, or the decision to submit the work for publication.

### Author contributions

Jenna A Moore-Ott, Data curation, Formal analysis, Investigation, Methodology, Software, Validation, Visualization, Writing – original draft, Writing – review and editing; Selena Chiu, Data curation, Investigation, Software; Daniel B Amchin, Methodology, Software; Tapomoy Bhattacharjee, Methodology; Sujit S Datta, Conceptualization, Formal analysis, Funding acquisition, Investigation, Methodology, Project administration, Resources, Supervision, Visualization, Writing – original draft, Writing – review and editing

## Author ORCIDs

Jenna A Moore-Ott http://orcid.org/0000-0001-6832-0658
Sujit S Datta http://orcid.org/0000-0003-2400-1561

## Decision letter and Author response

Decision letter https://doi.org/10.7554/eLife.76380.sa1
Author response https://doi.org/10.7554/eLife.76380.sa2

## Additional files

### Supplementary files

• Transparent reporting form

• Source data 1. Data for *Figure 4C*: Excel book summarizing the inputs used for each of the ten simulations shown in *Figure 4C* (i–vi). Different sheets within the book correspond to the individual bar graphs—Sheet "Bar Graph (i)" corresponds to all ten runs shown in (i), Sheet "Bar Graph (ii)" corresponds to all ten runs shown in (ii), and this follows for the remaining sheets. The first column identifies the input, its variable, and its units. The following columns correspond to each of the ten runs shown in the bar graph.

• Supplementary file 1. Ranges of the values of the parameters explored in our model with corresponding references.

• Supplementary file 2. Specific parameter values for the simulations shown in *Figures 1–3*, *Figure 2—figure supplement 1* and *Figure 3—figure supplement 1*.

### Data availability

All data generated or analyzed during this study are included in the manuscript and supporting file; source data files have been provided for all figures.

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

## Appendix 1

### Nondimensionalizing the governing equations

The governing equations *Equations 1–4* are described by six variables: those describing the concentrations of planktonic bacteria ($b_1$), biofilm bacteria ($b_2$), nutrient ($c$), autoinducer molecules ($a$), as well as the one-dimensional space ($x$), and time ($t$) coordinates. Additional constants for our equations are highlighted in *Supplementary file 1*, with initial conditions $b_1(t=0) = b_{1,0}$, $c(t=0) = c_0$, and $x_0$ as the width of the initial planktonic inoculum. We define the dimensionless variables $\tilde{b}_1 \equiv \frac{b_1}{B_1}$, $\tilde{b}_2 \equiv \frac{b_2}{B_2}$, $\tilde{c} \equiv \frac{c}{C}$, $\tilde{a} \equiv \frac{a}{A}$, $\tilde{x} \equiv \frac{x}{\mathcal{X}}$, and $\tilde{t} \equiv \frac{t}{T}$, where the tilde () notation indicates a dimensionless quantity and the dimensional quantities $B_1$, $B_2$, $C$, $A$, $\mathcal{X}$, and $T$ are not specified a priori. Thus, in nondimensional form, *Equations 1–4* can be represented as:

$$\text{Planktonic}: \frac{\partial \tilde{b}_1}{\partial \tilde{t}} = \frac{D_1}{(\mathcal{X}^2/T)} \tilde{\nabla}^2 \tilde{b}_1 - \frac{\chi_1}{(\mathcal{X}^2/T)} \tilde{\nabla} \cdot \left( \tilde{b}_1 \tilde{\nabla} \log \left( \frac{1+\tilde{c}/\tilde{c}_-}{1+\tilde{c}/\tilde{c}_+} \right) \right)$$
$$+ \tilde{b}_1(\gamma_1 T)g(\tilde{c}) - \tilde{b}_1(\tau^{-1}T)\mathcal{H}\left(\tilde{a} - a^*/A\right) \tag{S1}$$

$$\text{Biofilm}: \frac{\partial \tilde{b}_2}{\partial \tilde{t}} = (\gamma_2 T)\tilde{b}_2 g(\tilde{c}) + (B_1/B_2)(\tau^{-1}T)\tilde{b}_1 \mathcal{H}\left(\tilde{a} - a^*/A\right) \tag{S2}$$

$$\text{Nutrient}: \frac{\partial \tilde{c}}{\partial \tilde{t}} = \frac{D_c}{(\mathcal{X}^2/T)} \tilde{\nabla}^2 \tilde{c} - \left( (B_1/\mathcal{C})\kappa_1 T\tilde{b}_1 + (B_2/\mathcal{C})\kappa_2 T\tilde{b}_2 \right) g(\tilde{c}) \tag{S3}$$

$$\mathcal{X}^2/D_1, \mathcal{X}^2/chi_1, \mathcal{X}^2/D_c, \mathcal{X}^2/D_a\gamma_1^{-1}, \gamma_2^{-1}, \tau, \frac{c_o}{b_{1,0}k_1}, \frac{c_o}{b_{1,0}k_2}, \frac{a^*}{b_{1,0}k_1}, \frac{a^*}{b_{1,0}k_2}, \lambda^{-1}, (\alpha\beta_{1,0})^{-1} \tag{S4}$$

where $g(\tilde{c}) \equiv \frac{\tilde{c}}{\tilde{c}+\tilde{c}_{\text{char}}}$. Given that the characteristic autoinducer concentration $a^*$ arises in the argument of the Heaviside step function in *Equations S1 and S2*, we choose $A = a^*$. Moreover, given that the planktonic cells have a characteristic concentration $b_{1,0}$ defined by the initial inoculum, we choose $B_1 = b_{1,0}$. The fraction of the population in the biofilm state is defined as $f = b_2/(b_2 + b_1)$; thus, to ensure that $\tilde{f} = f$ for simplicity, we also choose $B_2 = B_1 = b_{1,0}$. Finally, given that the nutrient has a characteristic concentration $c_0$ defined by the initial saturation, we choose $\mathcal{C} = c_0$. With these choices of characteristic quantities, multiple length and time scales emerge as possible choices for $\mathcal{X}$ and $T$, respectively:

Length scale: $\sqrt{TD_1}$, $\sqrt{TD_a}$, $\sqrt{TD_c}$, $\sqrt{TD_a}$

Time scale: $\mathcal{X}^2/D_1, \mathcal{X}^2/chi_1, \mathcal{X}^2/D_c, \mathcal{X}^2/D_a, \gamma_1^{-1}, \gamma_2^{-1}, \tau,$
$\frac{c_o}{b_{1,0}k_1}, \frac{c_o}{b_{1,0}k_2}, \frac{a^*}{b_{1,0}k_1}, \frac{a^*}{b_{1,0}k_2}, \lambda^{-1}, (\alpha\beta_{1,0})^{-1}$

Each such choice will lead to the emergence of many different dimensionless groups characterizing this problem. Nevertheless, all these different groupings are accounted for in the dimensionless parameters $\tilde{\mathcal{D}}$, $\tilde{\mathcal{J}}$, and $\tilde{\mathcal{S}}$ described in the main text, with the exception of quantities involving the nutrient diffusivity $D_c$, planktonic-to-biofilm transition rate $\tau^{-1}$, and the natural autoinducer degradation rate $\lambda$, which have corresponding time scales that are much smaller than the other time scales of the systems considered here and are neglected from our analysis for simplicity.

## Appendix 2

### Derivation of the dimensionless parameters $\tilde{\mathcal{D}}$ and $\tilde{\mathcal{S}}$

We first estimate the time $\tau_d$ taken for cells to deplete available nutrient through consumption. To do so, for simplicity, we consider a population of planktonic cells exponentially growing at the maximal rate $\gamma_1$, uniformly distributed in a well-mixed and fixed domain (i.e., neglecting motility-mediated spreading), and consuming nutrient at the maximal rate $\kappa_1$. Thus, $\frac{dc}{dt} = -\kappa_1 b_{1,0} e^{t\gamma_1}$; integrating this equation from $t = 0$ (with $c = c_0$) to $t = \tau_d$ (with $c = 0$) yields *Equation 5* of the main text.

We use a similar approach to estimate the time $\tau_a$ taken for produced autoinducer to reach the threshold for biofilm formation $a^*$. In particular, we consider the same population of planktonic cells secreting autoinducer at the maximal rate $k_1$. We neglect natural degradation of autoinducer, given that the degradation rate is relatively small compared to binding to the cell surface receptors with a second-order rate constant $\alpha$, that is, $\lambda \ll \alpha b_0$. The rate of autoinducer production and loss are then given by $b_{1,0} e^{t\gamma_1} \times k_1$ and $b_{1,0} e^{t\gamma_1} \times \alpha a$, respectively, ultimately yielding $\frac{da}{dt} = b_{1,0} e^{t\gamma_1}(k_1 - \alpha a)$. Integrating this equation from $t = 0$ (with $a = 0$) to $t = \tau_a$ (with $a = a^*$) then yields *Equation 6* of the main text. Notably, this analytical solution for the time scale $\tau_a$ is only defined for $\tilde{\eta} \equiv \alpha a^* / k_1 < 1$; when $\tilde{\eta} \geq 1$, the rate of autoinducer loss exceeds that of autoinducer production and secretion, and thus the time required to reach the threshold for biofilm formation diverges. Finally, the ratio of $\tau_d$ and $\tau_a$ thus derived yields the nutrient availability parameter $\tilde{\mathcal{D}}$ as described in the main text.

Thus far, we have only considered nutrient consumption by planktonic bacteria. However, cellular proliferation, autoinducer production, and nutrient consumption can also occur for cells after they have transitioned to the biofilm state, causing biofilm-produced autoinducer to also drive surrounding planktonic cells to transition to the biofilm state. Hence, we repeat the same calculations for $\tau_a$ and $\tau_d$ as described above, but now for a population of cells in the biofilm state (still with the characteristic concentration $b_{1,0}$ defined in our model), exponentially growing at the maximal rate $\gamma_2$ and consuming nutrient at the maximal rate $\kappa_2$. In this case, $\frac{dc}{dt} = -\kappa_2 b_{1,0} e^{t\gamma_2}$, and integrating this equation from $t = 0$ (with $c = c_0$) to $t = \tau_{d,2}$ (with $c = 0$) yields $\tau_{d,2} = \gamma_2^{-1} \ln(1 + \tilde{\beta}_{2,0})$, where $\tilde{\beta}_{2,0} \equiv \gamma_2 / (b_{1,0} \kappa_2 / c_0)$ describes the yield of new biofilm cells produced as the population consumes nutrient. For the calculation of autoinducer production, we adopt a similar approach as that described above to calculate $\tau_a$, but now assuming that the biofilm surface receptors are saturated (i.e., neglecting autoinducer loss). As a result, $\frac{da}{dt} = b_{1,0} k_2 e^{t\gamma_2}$. Integrating this equation from $t = 0$ (with $a = 0$) to $t = \tau_{a,2}$ (with $a = a^*$) finally yields $\tau_{a,2} = \gamma_2^{-1} \ln\left(1 + \tilde{\theta}_{2,0}\right)$, where $\tilde{\theta}_{2,0} \equiv \frac{\gamma_2}{b_{1,0} k_2 / a^*}$. The ratio of $\tau_{d,2}$ and $\tau_{a,2}$ thus derived then yields the parameter $\tilde{\mathcal{S}}$ as described in the main text.

