## [Editor Report]

In this work, the authors develop a continuum description of biofilm formation from initially planktonic cells. The coupled partial differential equations that encode the dynamics of the cell populations, nutrients and autoinducers contain many parameters, but it is shown that only two dimensionless combinations of them are needed to understand the threshold for biofilm formation. This work should be of broad interest to a wide range of researchers in biophysics and cell biology.

---

## [Decision Letter]

**Decision letter after peer review:**

Thank you for submitting your article "A biophysical threshold for biofilm formation" for consideration by *eLife*. Your article has been reviewed by 2 peer reviewers, one of whom is a member of our Board of Reviewing Editors, and the evaluation has been overseen by Aleksandra Walczak as the Senior Editor. The reviewers have opted to remain anonymous. We regret the lengthy delay in furnishing this report.

Essential revisions:

1) Given that the continuum equations are by definition evolution dynamics for the mean values of the various quantities, are there situations in which a more probabilistic approach (e.g. by a Fokker-Planck equation) ought to be considered?

2) The problem studied here is simplified to one spatial dimension. How might the conclusions change if there were variations orthogonal to the main front propagation direction? Could this lead to qualitatively different conclusions?

3) Please clarify the use of a single effective diffusivity parameter for a system that is slowly being arrested. That is, is the diffusion coefficient a constant in equation 1? The effective diffusivity of cells changes dramatically from the planktonic to the immobilized state. How is this issue resolved in the model?

4) Biofilms are usually formed by different types of micro-organisms, and I am struggling to see whether this single species model is general enough to capture that reality. We do not mean to ask for the authors to perform more simulations, but we would like to hear know their thoughts on this issue.

5) While we believe that the results are compelling and quite useful, the work can be strengthened. For example, biofilms are usually heterogeneous (i.e., different microbial populations) and many of the parameters in the model are homogeneous. It would be interesting to understand how even a second cell species would alter (or not) the results, in particular the claim that onset of biofilm formation can be uniquely defined by the parameters D and J.

6) Are these results sensitive to the chosen initial condition? That is, it would be good to explore the dynamics of the biophysical model (perhaps that has been done already, but we did not see it in the manuscript). For example, are there oscillatory solutions or closed orbits, or even singularities that arise from the solution of the equations?*Reviewer #1 (Recommendations for the authors):*

The problem of biofilm formation is considered here in the context of a well-defined geometry (rectilinear) in which there are spatio-temporally evolving populations of planktonic and surface-bound cells in the biofilm, consuming nutrients, dividing and responding to secreted autoinducers. While there are many individual parameters that describe the various processes involved, including diffusion constants, motilities, growth, consumption, and transitions rates, the main result of the paper is that only two dimensionless combinations of parameters determine the threshold for biofilm formation. These are a ratio of timescales for nutrient depletion by planktonic cells and that for autoinducer to reach threshold defines one parameter, while the second is a ratio of times associated with the moving front of transitioning cells and the autoinducer timescale. Investigating a very wide range of the individual parameters in the problem shows that these two composite ones serve as a reliable predictor of biofilm formation.

This is an excellent paper that provides real insight into a notoriously difficult problem in biology. While the individual assumptions of the model can certainly be tweaked or generalised, the overall framework, and in particular the simplified geometry, should have a great impact on the field.*Reviewer #2 (Recommendations for the authors):*

The authors investigate the onset of biofilm formation (from a planktonic cell state) using a set of coupled, nonlinear differential equations. The equations model the two main cell states, planktonic and biofilm; two auxiliary equations for nutrient consumption and autoinducer production are introduced to model the transition from planktonic to biofilm state. Analysis of the biophysical model shows that there are two main dimensionless parameters that govern the onset of biofilm formation (in the model), namely a nutrient (D) and an autoinducer (J) availability parameters. An important result is that the biophysical threshold for biofilm formation seems quite robust across many different sets of model parameters suggesting, perhaps, a universal threshold. A strength of the formulation is that these dimensionless parameters can be experimentally measured, and therefore the presented biophysical model can be tested in the laboratory. Overall, the results from the model support the authors' claims.

---

## [Author Response]

Essential revisions:1) Given that the continuum equations are by definition evolution dynamics for the mean values of the various quantities, are there situations in which a more probabilistic approach (e.g. by a Fokker-Planck equation) ought to be considered?

The Reviewers and Editors raise an excellent point, and we thank them for this constructive suggestion. Our (deterministic) framework provides a first step toward modeling the process of biofilm formation. However, we fully agree that a probabilistic approach that can additionally describe e.g., stochastic variations in cellular behaviors would be a valuable extension of our work. Indeed, motivated by this insightful suggestion, we have started to explore how to incorporate such an approach into our model.

For example: our model describes the cellular processes of motility, nutrient consumption, proliferation, and autoinducer production, availability, and sensing using the parameters {D_1_,x_1_,c_,c_+_,k,c_char_,γ,k,α,τ,a*}, respectively—each of which is taken to be single-valued in each of our simulations. However, these parameters can have a distribution of values arising from e.g., inherent cell-to-cell variability in the underlying processes they describe. Because these values define the governing dimensionless parameters D~ and J~ that specify the threshold for biofilm formation in our model, we expect that variability in the parameter values would broaden the planktonic-to-biofilm transition predicted by our model. That is, we expect the transition specified by the black curve in Figure 4A to be smeared out, similar to what is seen in Figure 4D, though due to a fundamentally different reason—with biofilm formation arising in some cases at lower (D~, J~) than predicted by the black curve. Indeed, similar behavior was recently observed in a distinct model of biofilm formation on flat surfaces [Ref. 118 newly added to our manuscript]. Exploring the influence of such variations within our theoretical framework will thus be a useful direction for future research.

We appreciate the Reviewers and Editors for their insightful comment. In the revised manuscript, we have now explicitly discussed the possible ways in which a more probabilistic approach could be integrated with our model, and our expectations for how our results might be altered in such an approach, as described above. We anticipate that this extended discussion will help to motivate further work that builds on the present manuscript.

2) The problem studied here is simplified to one spatial dimension. How might the conclusions change if there were variations orthogonal to the main front propagation direction? Could this lead to qualitatively different conclusions?

We thank the Reviewers and Editors for this thoughtful question; such orthogonal variations could give rise to rich new effects that will be interesting to study in future work. Indeed, we examined a similar question in our prior work modeling the collective migration of planktonic bacteria in the *absence* of quorum sensing-mediated biofilm formation. In that case, we found that variations orthogonal to the main front propagation direction “smooth out” as the front propagates, due to corresponding variations in the chemotactic response of the cells; cells at outward-bulging parts of the front are exposed to more nutrients, which diminishes their ability to respond to the nutrient gradient via chemotaxis and thus slows them down [Refs. 96 and 97 of our manuscript]. As a result, the migrating front eventually smooths to a flat shape whose subsequent dynamics can then be described using just one spatial dimension, just as in our treatment in the present manuscript.

However, we expect that this behavior could be altered in interesting new ways when the cells can additionally produce and sense autoinducer and thereby transition to the biofilm state, as is the case in the present manuscript. In this case, we speculate that because cells at outward-bulging parts of the front are exposed to more nutrients and have a weaker chemotactic response, autoinducer production and accumulation will be more rapid relative to cellular dispersal. That is, at these parts of the front, τ_α_ and τ_c_ will be shorter and longer, respectively, causing the dispersal parameter J~ to be larger locally. Thus, our model would predict biofilm formation to occur first at these parts of the front, potentially also influencing subsequent dispersal and biofilm formation at other locations along the front. Therefore, while the conclusions of our present manuscript could be the same *locally* at different parts of the front, the *global* behavior of the population could indeed be different—potentially giving rise to e.g., spatially-heterogeneous biofilm formation.

In the revised manuscript, we have now explicitly discussed possible ways in which variations orthogonal to the main front propagation direction could lead to fascinating new behaviors, as described above. We anticipate that this extended discussion will help to motivate further work exploring this direction of inquiry, and thank the Reviewers and Editors for encouraging us to think about these possibilities.

3) Please clarify the use of a single effective diffusivity parameter for a system that is slowly being arrested. That is, is the diffusion coefficient a constant in equation 1? The effective diffusivity of cells changes dramatically from the planktonic to the immobilized state. How is this issue resolved in the model?

We apologize for not adequately clarifying how the cellular diffusivity changes from the planktonic to the immobilized state in the previous version of the manuscript, and appreciate having the chance to do so here. In our model, the diffusivity of cells in the planktonic state (in Equation 1) is indeed a constant D_1_. When these cells encounter sufficiently concentrated autoinducer (a ≥ a^∗^), they transition to the immotile biofilm state after a time lag ~τ.

The Reviewers and Editors astutely point out that in real systems, the change in cellular diffusivity (and chemotactic coefficient) may not be as temporally abrupt—although we are not aware of specific characterization of these dynamics, which would be useful for future experiments to probe. Thus, in the absence of such characterization, for simplicity, we treat this transition as being step-like from a single constant value of diffusivity D_1_ and a single constant value of the chemotactic coefficient χ_1_ in the planktonic state to zero diffusivity and chemotactic coefficient in the biofilm state after the time lag ~τ. Future work could, for example, incorporate a temporally-varying diffusivity (and chemotactic coefficient) into our model that slowly transitions from the planktonic values D_1_ and χ_1_ to the biofilm value of zero over a non-zero time scale. Given that the same cells would be transitioning from the motile planktonic to immotile biofilm state—but in this case with the introduction of a time-varying diffusivity and chemotactic coefficient—we expect that the long-time biofilm fraction f will be similar, and only the spatial profile of the biofilm population may be altered. Hence, we expect that our main findings summarized in Figure 4 will be unaffected. To test this expectation, we have now run a new version of the identical representative simulation shown in Figure 1C, but with both motility parameters D_1_ and χ_1_ smoothly transitioning to zero as described above. We observe nearly-identical results for both cases, confirming our expectation that the temporal nature of the arrest in motility does not appreciably influence our model results and conclusions.

We thank the Reviewers and Editors for asking this interesting question. In the revised manuscript, we have now explicitly discussed possible ways in which these dynamics could be incorporated into our model, as described above, along with the new results presented above. Further investigating these dynamics will be an interesting direction to explore in future work.

4) Biofilms are usually formed by different types of micro-organisms, and I am struggling to see whether this single species model is general enough to capture that reality. We do not mean to ask for the authors to perform more simulations, but we would like to hear know their thoughts on this issue.5) While we believe that the results are compelling and quite useful, the work can be strengthened. For example, biofilms are usually heterogeneous (i.e., different microbial populations) and many of the parameters in the model are homogeneous. It would be interesting to understand how even a second cell species would alter (or not) the results, in particular the claim that onset of biofilm formation can be uniquely defined by the parameters D and J.

We are responding to both questions 4 and 5 together, because they are closely related. We are grateful to the Reviewers and Editors for encouraging us to clarify possible ways our model could be extended to consider variability in cellular behaviors in a population, as well as the inclusion of multiple species in a biofilm community. Indeed, we believe our theoretical framework provides a useful foundation for modeling these important features of real-life systems.

As the Reviewers and Editors astutely pointed out, biofilms are often formed by multiple different microbial species, which we do not consider in this paper. Instead, as a first step, our work describes biofilm formation by a single species, for simplicity. Nevertheless, we expect that our theoretical framework can be extended by following reasoning similar to that described in our paper, but with the introduction of additional equations and variables in the governing Equations (1)— (4) to describe the distinct cell and chemical types. For example, in the case of biofilm formation by two different species, A and B, under positive quorum sensing control:

i) In the case that the different species consume and respond to distinct nutrients c_i_ (where i ∈ {A,B} indexes each species), and secrete and respond to distinct autoinducers a_i_, each species can be described in isolation using our same governing Equations (1)—(4), but now extended to incorporate the distinct variables c_i_, a ,b_1,i_, and b_2,i-_. Then, directly following the approach described in our paper, each species would be described by its own dimensionless parameters Di~ ≡ Td, i/Ta, i and Ji~ ≡ Tc, i/Ta, i, with D∗~/D~ +J∗~/J~ ∼  1 again specifying the threshold for biofilm formation for each. We hypothesize that the composition of the final two-species biofilm community would then be given by the combination of each single-species biofilm.

ii) In the case that the different species consume and respond to the same nutrient c, and secrete and respond to the same autoinducer +, our governing Equations (1)—(4) could be extended to consider the cellular parameters specific to each species i. In this approach, however, biofilm formation by each of the two species cannot be described in isolation, because they are coupled through the nutrient and autoinducer dynamics. For example, the calculation of the nutrient depletion time scale τ_d_ would need to be extended—following the same derivation as in our paper—to now reflect the aggregate nutrient consumption by both species. Similarly, the time scales τ_a_ and τ_c_ would be extended, following our paper, to now reflect contributions from both species. Then, just as in our paper, the overall two-species community would be described by one set of governing dimensionless parameters (D~,J~) with D∗~/D~ +J∗~/J~~1 specifying the threshold for biofilm formation for the overall twospecies community. We hypothesize that while this relation would specify a universal biophysical threshold for the onset of biofilm formation by the entire community, the composition of the final two-species biofilm that results above this threshold may not be uniquely specified by (D~,J~).

The above serve as two possible examples of how our model could be extended for the case of two-species communities. Similar extensions can be made as needed for e.g., communities with even more species, or communities with species that transition to/from the biofilm state in different ways (e.g., reversible biofilm formation, biofilm formation under negative quorum sensing control or regulated by other non-quorum sensing-based mechanisms).

We thank the Reviewers and Editors for encouraging us to think about the possible ways our work could be extended to describe additional complexities of biofilm formation in real-life settings. In the revised manuscript, we have now explicitly discussed the simplifications made in our model, its limitations, and the possible ways in which it could be extended to describe different cell types and mechanisms of chemotaxis/biofilm formation. We anticipate that our model will provide a foundation for other researchers to theoretically describe these and other complexities.

6) Are these results sensitive to the chosen initial condition? That is, it would be good to explore the dynamics of the biophysical model (perhaps that has been done already, but we did not see it in the manuscript). For example, are there oscillatory solutions or closed orbits, or even singularities that arise from the solution of the equations?

We thank the Reviewers and Editors for these interesting questions, and appreciate having the chance to clarify the details of our model and simulations. Because the goal of this present manuscript is to present the biophysical model and examine the predictions it makes for the onset of biofilm formation, our work thus far used numerical solutions of the full system of equations. So, we have not yet fully probed the mathematical structure of the governing equations to examine the possibility of e.g., oscillatory solutions, closed orbits, or singularities—which would be fascinating to explore in the future. We expect that oscillatory solutions/closed orbits are not likely to arise, given that our model takes biofilm formation to be irreversible; however, such complex long-time dynamics may emerge if the transition to biofilm formation were taken to be reversible. More broadly, exploring our system of equations within the framework of dynamical systems theory will be an interesting direction for future work.

However, to assess the sensitivity of our results to the chosen initial condition, we performed a new simulation identical to that shown in Figure 1C, but with the shape of the initial inoculum changed from a Gaussian profile to a step function (with the same maximum cellular concentration and width). We observe nearly-identical results for both cases, indicating that the results are robust to variations in this initial condition chosen. The influence of other changes in the initial conditions (e.g., maximal bacterial concentration, nutrient concentration) is already described by our model through the time scales τ_d,_τ_a_, and τ_c_ that define the governing dimensionless parameters (D~,J~).